# Comparison of a Group-/Home-Based and a Weight-Machine-Based Exercise Training for Patients with Hip or Knee Osteoarthritis—A Secondary Analysis of Two Trial Interventions in a Real-World Context

**DOI:** 10.3390/ijerph192417088

**Published:** 2022-12-19

**Authors:** Inka Roesel, Inga Krauss, Peter Martus, Benjamin Steinhilber, Gerhard Mueller

**Affiliations:** 1Institute for Clinical Epidemiology and Applied Biostatistics, University Hospital of Tuebingen, 72076 Tuebingen, Germany; 2Department of Sports Medicine, University Hospital, Medical Clinic, 72076 Tuebingen, Germany; 3Interfaculty Research Institute for Sports and Physical Activity Tuebingen, 72074 Tuebingen, Germany; 4Institute for Occupational and Social Medicine and Health Services Research, University Hospital of Tuebingen, 72074 Tuebingen, Germany; 5Allgemeine Ortskrankenkasse AOK Baden-Wuerttemberg, 70191 Stuttgart, Germany

**Keywords:** hip osteoarthritis, knee osteoarthritis, group training, weight-machine-supported training, health care research, WOMAC, short-term results

## Abstract

This study aimed to compare an individual weight-machine-based strengthening program (MbT) with a group-/homebased training offering strengthening/functional exercises (GHT) in a general health care setting. A total of 657 participants (GHT = 521, MbT = 136) suffering from hip/knee OA were included and analysed with a pre–post design (baseline (T0)/3-months (T1)). Primary outcomes were pain and physical functioning (Western Ontario and McMaster Universities Osteoarthritis Index, range 0–10). Additionally, adherence and perceived patient benefit were measured (T1). Data were analysed with linear mixed models (time, treatment, baseline pain/physical impairment severity) adjusted for patient characteristics. No significant between-group differences in pain reduction/functional improvements (time*treatment*baseline pain/physical impairment severity, pain/function: n.s.; time*treatment, pain: *p* = 0.884, function: *p* = 0.067). Within-group improvements were dependent on baseline severity: Higher severity levels demonstrated larger changes from baseline. Perceived patient-benefit (very high to high, GHT: 78%, MbT: 92%) and exercise adherence (Dropouts T1: GHT: 27.8%, MbT: 16.2%; adherence to supervised sessions: GHT: 89%, MbT: 92%) was slightly better in the MbT. In summary, both MbT and GHT, showed positive results for patients with at least moderate disease symptoms. Findings for physical functioning, perceived patient-benefit, exercise adherence hint towards a superiority of MbT. Individual preferences should be considered when prescribing exercise therapy. Trial registration: (1) German Clinical Trial Register DRKS00009251. Registered 10 September 2015. (2) German Clinical Trial Register DRKS00009257. Registered 11 September 2015.

## 1. Introduction

Osteoarthritis (OA) is a degenerative disease which predominantly affects weight-bearing joints such as the hip and knee. OA ranks among the ten most disabling diseases in high-income countries and is a major cause of pain, joint stiffness, and limitation in physical function [1]. Amongst other factors, obesity, muscular deficits and disbalances in the lower limbs/trunk causing abnormal mechanical joint forces and poor movement quality have been related to the development and progression of OA [2].

Exercise training can counteract these factors and is widely recommended by the current body of research as a key pillar in first-line conservative non-pharmacological treatment to manage the symptoms of knee and hip OA and achieve pain relief and improved physical functioning of the joints [3,4,5,6]. However, exercise interventions can differ greatly with respect to exercise type (e.g., strengthening, flexibility, neuro-motor skills or mixed), mode of resistance used for strength training (i.e., training devices), dosage principles (e.g., frequency), and delivery format (e.g., individual vs. group-/home-based) [3,7,8,9,10]. So far, no conclusive recommendation on the most favourable exercise regime can be generated from previous study findings, as head-to-head trials comparing different training settings are mostly very specific to the endpoint and the population under study [11,12], or vary in many components with respect to the exercise modalities, so that it remains unclear which differences between the training programs are the relevant factors influencing the outcome [3].

Recent meta-analyses suggest that land-based exercises combining different types of exercises (mixed training) are less effective than strength training alone [7,13]. For strengthening exercises, there is no clear evidence suggesting that treatment benefits significantly vary with the specific type of resistance training [14]: Using therapeutic elastic bands or using weight-machines can be equally effective. However, it remains an open question whether dosage can be appropriately applied using therapeutic elastic bands, as the level of resistance is difficult to control and to measure [3]. Establishing recommendations on optimal dosage of frequency, intensity, and exercise duration, specifically for patients with hip or knee OA, is also challenging as most trials vary in multiple dosage parameters at the same time [3,8,10,14,15]. Beneficial effects have been reported, specifically for alleviation of OA symptoms, if interventions comprise 12 or more supervised exercise sessions [8]. This finding is in line with the general recommendation that exercises should initially be instructed under close supervision [5,16]. In this regard, individually delivered programs may be preferable in comparison to group-based or home-based exercises [8].

Despite these hints towards an optimized exercise regime for patients with OA, the most effective format remains an open field for research. Besides the high economic burden of OA to health systems, it is of utmost importance to health care providers for ethical reasons to attain a deeper knowledge about the most beneficial and at the same time most feasible exercise intervention to guarantee a patient-oriented care. This includes, among other aspects, an individualized treatment according to the wishes and expectations of the individual and OA-related risk factors and disease severity [16,17].

The study presented here contributes to this ongoing research and evaluates two different exercise interventions offered to participants diagnosed with hip and/or knee OA:(a)a group-/home-based hip and knee training (GHT)–*targeting muscle strength, flexibility, motor learning and postural control (mixed training type)*.(b)an individual weight-machine-based hip and knee training (MbT)–*targeting muscle strength (strength training type)*.

Both training programs were implemented by a health care provider in a real-word health-care setting. The exercise interventions aimed at quite similar physiological loads, however, they differed with respect to exercise type, number, and delivery mode of supervised sessions, as well as to training devices. Our objective was to directly compare the short-term results (3-month-post-intervention) of the two training programs with respect to pain reduction and functional improvements. We hypothesized MbT to be superior to GHT because of exercise specifics described above anticipating a better outcome. We furthermore considered different initial degrees of pain and functional impairments and explored how individual characteristics such as age, sex, BMI and site of OA might affect the outcomes and whether the patients’ individual preferences might need to be considered when prescribing the potentially best exercise program [3,13,18,19].

## 2. Materials and Methods

Parts of this section correspond to the previously published study protocols of the two exercise interventions under study [20,21].

### 2.1. Study Design and Setting

This study is a comparative analysis of the intervention groups of two separate prospective multi-centre non-randomized controlled trials on exercise interventions in subjects with hip and/or knee OA. Both trials were set in a real-world scenario and represent pragmatic trials in the context of health services research. They were conducted in collaboration between a statutory health insurance company in the federal state of Baden-Wuerttemberg in Germany (Allgemeine Ortskrankenkasse, AOK, Baden-Wuerttemberg, Stuttgart, Germany) and the University Hospital of Tuebingen, Tuebingen, Germany (UKT).

A pre–post design was used in the present study: Primary outcomes were assessed prior (T0 = pre) and immediately after (T1 = post) the intervention period for GHT (11-weeks) and after the first 12-week intervention phase for MbT. As treatment exposure is evident, blinding of participants and health professionals was not possible. Data analysts were not blinded either.

### 2.2. Participant Eligibility and Recruitment

The group-/home-based hip and knee training (GHT) was provided at more than 70 AOK health service centres, the weight-machine-based training (MbT) was offered at two of those centres as a pilot project. Prerequisites for participation in both exercise interventions were: (1) AOK membership, (2) referral from an orthopaedic specialist/general practitioner due to hip and/or knee complaints, (3) absence of comorbidities putting the patient at risk when exercising. Eligible customers were allocated to the AOK centres nearest to their place of residence. At the two study sites offering GHT as well as MbT, participants made a joint decision with a health care professional on which intervention to choose. Subscribers to the training programmes were then requested to participate in the accompanying scientific evaluation. In case of a positive response, they received postal mail including the study information sheet and a paper–pencil questionnaire asking for further in- and exclusion criteria and a confirmation of a (self-reported) lifetime prevalence of hip and/or knee OA diagnosed by a medical practitioner.

More detailed in and exclusion criteria are outlined in Appendix A.

### 2.3. Interventions

Exercise instructions were provided by specially trained and qualified exercise professionals of the insurance company at the respective study centres.

Table 1 gives an overview of the most relevant aspects of the interventions. While both interventions were progressive training programmes with dosage parameters according to ACSM recommendations [22], the interventions differed with respect to (a) the type of training, (b) type of resistance, (c) mode of delivery and (d) the number of sessions:(a)Both programmes had a major focus on strengthening exercises. However, the MbT complemented these exercises with stretching exercises for the muscles in charge only, whereas GHT also comprised specific exercises to improve flexibility, motor skills and postural control. MbT can therefore be classified as a strength training programme, GHT rather as a mixed training programme including relevant strengthening components.(b)MbT used weight-machines in the first training set followed by another set using small training devices or functional exercises applying body weight as resistance. The latter were the only types of resistance used for the GHT.(c)Participants of MbT received individual training with a personal 1:1 supervision for the first three visits and were then monitored in groups of three to ten people for the subsequent exercise sessions; delivery format for the GHT was group-based with additional unsupervised home-exercises.(d)The number of supervised exercise sessions (8 GHT, 24 MbT) and the total number of scheduled training sessions (30 GHT, 24 MbT) differed.

Detailed information on the exercise regimes can be found in the study protocols of the original trials [20,21].

### 2.4. Outcomes

***Primary Outcomes: WOMAC Pain and Function.*** The Western Ontario and McMaster Universities Osteoarthritis Index (WOMAC^®^ NRS 3.1 German Index, Department of Rheumatology, University Hospital Zurich, Zurich, Switzerland) is a widely used disease-specific, validated, and reliable questionnaire to measure self-reported symptoms and physical disability in individuals with osteoarthritis (OA) of the hip and/or knee [23]. Short-term results of the two exercise interventions under study were compared with the WOMAC subscales pain and physical function [0 = no limitation; 10 = worst limitation]. All outcome measures were patient-reported.

***Perceived patient benefit from the intervention***. The participants’ perceived benefit from the interventions was assessed on a 5-point Likert scale (1 = no perceived benefit to 5 = very high perceived benefit).

***Exercise adherence.*** The number of attended training sessions was self-reported at T1.

***Covariates.*** Age, sex, body mass index (BMI), site of OA (hip/knee/both), and additional joint replacement (yes/no) were reported at baseline.

### 2.5. Sample Size

As the two intervention groups originated from the two separate trials with distinct sample size calculations [20,21], no additional sample size estimation was conducted for this secondary analysis.

### 2.6. Statistical Analysis

According to the Intention-to-Treat (ITT) principle, the presented analysis was applied to subjects who provided data at baseline in the primary endpoints under the assumption of a missing at random (MAR) mechanism. A total of two separate linear mixed models (LMMs) with a random intercept for participants were conducted for the primary outcomes WOMAC pain/function. Model assumptions were adequately fulfilled.

As the baseline disease status can influence response to treatment [24], we aimed to assess the correlation between random intercept and slope, however, LMMs with random slope led to non-convergence due to insufficient observations to support the corresponding models. Instead, we introduced baseline disease severity subgroups of WOMAC pain and function, categorizing severity into low (1st tercile; pain < 2, function < 1.65), medium (2nd tercile; pain [2; 3.8], function [1.65; 3.35]) and high (3rd tercile; pain > 3.8, function > 3.35) to test for moderating effects of initial disease severity on the relationship between intervention group and pain/function over time. The WOMAC tercile cut-offs were chosen exploratively based on a study by Weigl, Angst [25].

Models (1p: pain, 1f: function) incorporated treatment, time, baseline disease severity and the according interaction terms. Furthermore, they were adjusted for confounding covariates (see Section 2.4), which were treated as fixed effects. Interaction effects of the covariates with time and treatment were investigated, yet not significant, thus not included in the final models. The significance level was set at 0.025 (two-tailed, Bonferroni correction) for the two primary outcomes. Estimated marginal means (EMMs) of the LMMs and the corresponding predicted mean change from baseline values (cfb) were calculated.

All data analyses were conducted with the statistical software R version 4.0.3, R Foundation for Statistical Computing, Vienna, Austria [26], and IBM SPSS Statistics, version 27, IBM Corp., Armonk, NY, USA.

## 3. Results

### 3.1. Participants

#### 3.1.1. Recruitment and Participant Flow

Participants for GHT and MbT were recruited between September 2015 and April 2017. Details on participant flow are provided in Figure 1. Finally, 657 participants (GHT = 521, MbT = 136) could be included in our analyses.

#### 3.1.2. Dropouts

The overall rate of participants who prematurely dropped out of the study was 12.0% (n = 79) of the sample. In the GHT 12.5% (n = 65) and in the MbT 10.3% (n = 14) of the participants were lost to follow-up. We assessed factors related to drop-out from the study by comparing completers (n = 587; 88.0%) and dropouts on all studied variables (Appendix A). We found a statistically significant association between sex and study completion (*p* = 0.044), with dropouts being more frequently female. Dropouts furthermore exhibited significantly higher baseline WOMAC pain (*p* < 0.001) and function scores (*p* < 0.001).

#### 3.1.3. Baseline Characteristics

Participant characteristics at T1 within the two groups are displayed in Table 2.

The majority of the study population was female. Age of the total study population ranged from 22 to 90 years. The sex proportion was significantly different in the two intervention groups with a balanced gender ratio in the MbT, whereas in the GHT over three quarters of the participants were females. A significant difference in age was found between the two intervention groups. No significant differences in the two groups were observed in BMI and joint anamnesis.

### 3.2. Primary Outcomes: WOMAC Pain and Function

Descriptions of the primary outcome scores at T0 and T1 are displayed in Table 3.

Baseline (T0) WOMAC pain and function scores were not significantly different among the two intervention groups (pain: *p* = 0.567, function: *p* = 0.452).

Results of the linear mixed models for WOMAC pain and function are displayed in Table 4. Neither of the two LMMs revealed significant *time***treatment*baseline disease severity* interactions (*Model 1p*: *p* = 0.945, *Model 1f*: *p* = 0.209). Also, the *time*treatment* interaction for pain was non-significant (*Model 1p*: *p* = 0.884) and failed to reach statistical significance for WOMAC function (*Model 1f*: *p* = 0.067). The *time*baseline disease severity* interactions were found to be significant for both models (*p* < 0.001).

Figure 2 shows the estimated marginal means (SE) of the WOMAC pain and function scores at baseline (T0) and post-intervention (T1) stratified by baseline severity (see also Table 5).

Whereas slight, yet non-significant deteriorations in WOMAC pain levels from baseline to T1 were found in the category of low initial pain severity (MbT: cfb = 0.117, *p* = 0.999; GHT: cfb = 0.102, *p* = 0.999), significant improvements over time were demonstrated in the categories of medium (MbT: cfb = −0.520, *p* = 0.024; GHT: cfb = −0.488, *p* < 0.001) and high baseline pain levels (MbT: cfb = −1.192, *p* < 0.001; GHT: cfb = −1.268, *p* < 0.001) for both interventions.

With respect to WOMAC function, participants with a low baseline functional impairment exhibited a minor non-significant worsening of symptoms in both exercise groups (MbT: cfb = 0.208, *p* = 0.999; GHT: cfb = 0.102, *p* = 0.999). In the category of medium initial functional limitations, physical functioning improved significantly in subjects of the MbT group (cfb = −0.466, *p* = 0.016), whereas change from baseline in the GHT was non-significant (cbf = −0.094, *p* = 0.999). Analogous to the findings for WOMAC pain, participants with the highest baseline impairments experienced the largest functional improvements (MbT: cfb = −1.192, *p* < 0.001; GHT: cfb = −1.268, *p* < 0.001).

Detailed model results for WOMAC pain and function with respect to the fixed effects of the included covariates can be found in Appendix A. None of the interactions of the covariates with time and treatment yielded statistical significance, the covariates neither had a moderating effect on the time-treatment interaction, nor were they predictive of the change in scores over time, thus these interactions were excluded from the models.

Nevertheless, a higher age was significantly associated with generally higher WOMAC pain (β_age_ = 0.01, 95% CI: 0.00–0.02, *p* = 0.022) and worse WOMAC function scores (β_age_ = 0.01, 95% CI: 0.00–0.02, *p* = 0.020). A higher BMI was positively associated with greater functional impairments (β_age_ = 0.02, 95% CI: 0.01–0.04, *p* = 0.006). This association failed to reach statistical significance for WOMAC pain (*p* = 0.062).

### 3.3. Exercise Adherence

Overall, 474 patients (GHT: n = 370; MbT: n = 101) provided information on their attended training sessions. Exercise adherence was high in both intervention groups with an average of 92% attended sessions out of the 24 scheduled supervised sessions (mean = 22.2, SD = 3.2; median = 24, IQR = 2; n = 101) of the MbT. Discriminating between group (n = 8) and home sessions (n = 22) for the GHT, 89% of the supervised group sessions (mean = 7.1, SD = 1.2; median = 7, IQR = 1; n = 370) and 93% of the home sessions (mean = 20.6, SD = 3.9; median = 22, IQR = 0; n = 346) were completed.

### 3.4. Perceived Benefit from the Exercise Intervention

In total, 490 patients provided information on their perceived benefit from the interventions. Overall, 398 (81.2%) subjects stated a *very high to high* perceived benefit from the training programmes (GHT: n = 293, n = 77.9%; MbT: n = 105, 92.1%). *Some benefit* was reported by 17.0% (n = 64) in the GHT and 6.1% (n = 7) in the MbT group. Five percent of the participants in the GHT (n = 19) and 1.8% (n = 2) in the MbT indicated to derive only *small* or *no benefit* from the intervention. A statistically significant association between intervention group and perceived benefit in favour of MbT was found (Fischer’s exact test: *p* < 0.001). Spearman correlations between the perceived benefit and pre–post changes in WOMAC pain and function were low with r = 0.17 and r = 0.19, respectively.

### 3.5. Harms

No severe adverse events were reported to the principal investigators throughout the study.

## 4. Discussion

We evaluated the short-term results of two progressive training interventions in a real-world health care setting. The interventions differed with respect to the type of training, type of resistance, mode of delivery and the number of sessions, while the overall physical load can be regarded as similar. Our study therefore contributes to the body of research on the relative effectiveness of different exercise programs.

In a direct comparison of the two types of exercise interventions, we could not demonstrate a statistically significant difference in pain reduction or in functional improvements between GHT and MbT in the short-term. Our hypothesis of MbT being superior to GHT was mostly based on recent systematic reviews [7,13], which identified mixed exercise types as the least effective among other single-type (e.g., strength) training programs. Moreover, a close supervision of the patients during training sessions was recommended [5,16], which we supposed to be guaranteed in the MbT rather than in the GHT due to the direct 1:1 interaction with the health professional in the first three sessions and the subsequent possibility for individual feedback and advice while exercising in the gym.

Our findings opposed to our hypothesis might be partially attributed to the inconsistency of the definition of a mixed training program among different studies [3]. It is not possible to attribute greater weight to particular components of the respective mixed exercise programs in comparison with other programs, so it might be misleading to generally consider mixed programs as the least effective. Furthermore, a downside of mixed programs is the conflicting molecular response caused by resistance training and aerobic training within the same session [13]. As the GHT did not have focus on aerobic activities, this issue does not apply.

We may have also underestimated the intensive support during group sessions by health professionals who had been explicitly trained to respond to the specific needs of each participant. Training programs such as the GHT may further induce positive effects due to their social component and the dynamics of training in the group [11]. Our results are furthermore in accordance with Juhl, Christensen [13], who did not find statistically significant differences in subgroup analyses according to the number of supervised sessions.

Socio-demographics and OA-specific covariates such as age, sex, BMI, site of OA and artificial joint replacement were not found to exhibit an effect on change from baseline or to moderate the effect of the intervention type. However, within-group changes from baseline and effects over time were dependent on the initial baseline severity of functional limitations and pain: participants starting with higher initial disease status improved more by means of the interventions than subjects with a low baseline severity who even slightly worsened. Statistically significant within-group differences could only be demonstrated for subgroups of medium and high baseline severity of pain (MbT and GHT) and medium (MbT) and high baseline severity of function (MbT and GHT). The within-group changes for the high baseline severity groups were above the recommended cut-off values to determine the minimal clinically important difference (MCID), whereas the medium baseline severity groups were found to be smaller [27]. On the other hand, it has been reported that MCIDs are related to the baseline disease severity as well. Patients with only mild symptoms need less improvement to perceive a personal benefit, whereas with higher baseline symptom severity levels, change scores must be larger as well to be perceived as clinically important [28]. This finding may explain our results on perceived benefit from the intervention: Despite the rather small effects of both interventions for both outcomes in our study, most participants who completed the intervention were satisfied with the program, indicating a high to very high personal benefit. Further reasons for the positive evaluation of both interventions may be linked to the fact that perceived patient benefit is not only related to pain and functional outcomes, but also to factors relating to the instructor (personality, professionalism, motivational skills), program design (including location, exercise content, affordability) and social connectedness in case of group-based exercises [29]. Furthermore, a bias of results due to social desirability cannot be ruled out.

Despite the absence of statistically significant differences in pain and physical function between MbT and GHT, there are some aspects favouring the machine-based training. In view of perceived patient benefit, the proportion of patients stating a *high to very high* benefit from the training programs was significantly higher in the MbT (92%) than in the GHT (78%). Regarding adherence to the scheduled exercise sessions, one might have expected a stronger adherence to the group exercises than to the individual machine-based training sessions due to the social component and dynamics of training in a group [11]. Yet, adherence to both exercise regimens was very similar with even a slightly higher exercise compliance in the MbT (92% vs. GHT 89% (supervised sessions)). This may be due to the initial 1:1 supervision, when participants were introduced to the training dosage and the use of weight-machines. This personal interaction with the trainer could have fostered the motivation of the participants. Also, appointments could be scheduled as desired by the patient, whereas appointments for GHT were fixed.

We were also interested in individual characteristics of the participants taking part in either MbT or GHT and of those ceasing the study prematurely. Dropouts were more frequently female in comparison to the completers and were further characterized by more pain (GHT and MbT) and worse physical functioning (GHT) at baseline (Appendix A). This is in accordance with other studies reporting a lower self-rated baseline health and higher pre-exercise arthritis medication usage (as a surrogate marker of a worse baseline health as well as higher pain levels) before ceasing a study [30,31,32]. Although exercise is recommended for all patients with OA, increasing physical complaints or adverse events may therefore be a relevant barrier for exercise participation and intervention strategies such as land-based/aquatic non-weight-bearing exercises or mind-body exercises which integrate mindfulness and relaxation into physical movements (e.g., tai chi, yoga) could provide beneficial alternatives for patients not adhering to higher load exercise programs [7,19]. However, this hypothesis needs further exploration in the future.

Guidelines on exercise recommendation have acknowledged that “one-size-fits” all approaches might attenuate treatment effects and that certain sub-groups with specific characteristics need tailored approaches: Treatment of hip and/or knee OA should be individualized according to the wishes and expectations of the individual [16]. Our study adds to this discussion and underlines the potential advantages of providing both GHT and MbT interventions in the health care system from a person-oriented view: We revealed significantly different participant characteristics in the two intervention groups. Given the background of an approximately 1.5-fold higher prevalence rate for OA for women in comparison to men [33], it has to be noted that the overall percentage of female study participants in comparison to men exceeded this rate (ratio = 2.5:1). This surplus in women is even more pronounced in the GHT (ratio = 3.2:1), whereas almost half of the participants of the MbT were male, hinting towards a sex-related preference-based selection of the intervention program. This finding is in line with a previous study stating that social interaction as a relevant motive for exercise participation was more prominent in female participants [17]. We also found an age-related difference between participants of the GHT in comparison to the MbT, the latter being preferred by younger participants. These findings underline the necessity of an individualized OA treatment [16], considering a person’s characteristics and preferences with regard to the type and setting of exercise.

### Limitations

A limitation of this study is the lack of a non-exercise control group. Results observed in this study could represent effects of the intervention or a natural course over time. However, considering the progressive nature of OA, one would rather expect a worsening of the symptoms over time. Nonetheless, regression-to-the-mean effects cannot be ruled out, which might partially explain the strong baseline dependency of our results. Future research will compare the exercise interventions, GHT and MbT, with a non-intervention control group, allowing a more detailed statement on this issue.

Furthermore, detailed study population information on study site level were not available. Therefore, we do not know the characteristics of participants of the health care interventions who declined participation in the accompanying study (=patients who made use of the exercise programs but did not give consent to be included in this analysis). They may have differed from study subjects with respect to their personal attributes. Moreover, as a real choice between GHT and MbT was only offered at the two MbT-pilot study sites, a final statement on associations between patient characteristics and exercise intervention preferences cannot be made from this data.

Lastly, it is a drawback that the comparison of the two exercise interventions cannot be considered as a real experimental head-to-head trial, as the study was not originally planned as such und lacks conformity regarding frequency of sessions and duration of follow-up time. Nevertheless, exercise manuals for both training programs were designed by the same institution aiming for quite similar physiological loads and both interventions were conducted in a real-world scenario within the same target population. We furthermore did not conduct a sample-size calculation for the comparison of MbT versus GHT as it was a secondary explorative analysis of the intervention groups of two separate controlled trials, for which power calculations had been performed separately. Due to the unbalanced design and a potential lack of sufficient power our study might have failed to detect a differential effect for physical functioning (Time * Treatment, *p* = 0.067).

## 5. Conclusions

Our results suggest that exercise programmes in the context of community health services can improve the short-term course of pain und functionality regardless of the type and setting of the training regimens in patients with at least moderate OA symptoms.

Findings for physical functioning, patient-reported satisfaction with the training regimens and exercise compliance point towards a slight superiority of the machine-based training. Moreover, the two exercise modalities under study exhibited significantly different participant characteristics, which hints towards individual preferences and underlines the necessity of a person-oriented perspective in exercise interventions. This necessity is further supported by the fact that effects were dependent on the initial disease severity of a person.

Further research is needed whether exercise programmes specifically tailored to different patient subgroups can maximize the benefits of exercise interventions offered by health services.

## Figures and Tables

**Figure 1 ijerph-19-17088-f001:**
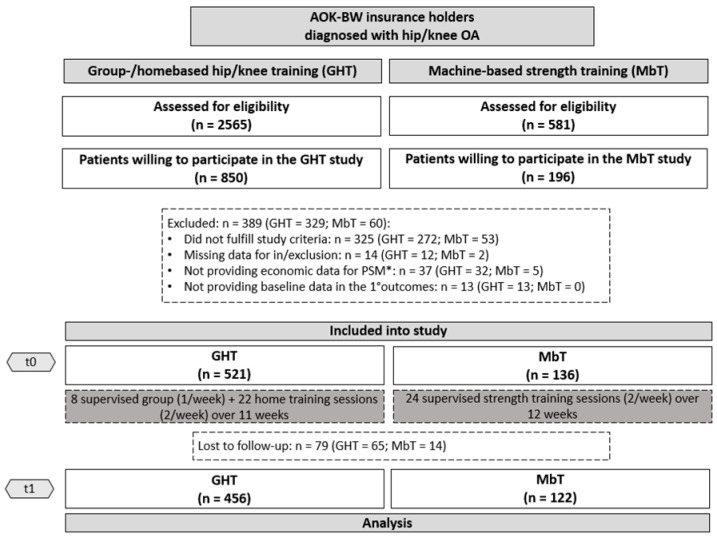
Flowchart. *PSM = Propensity score matching. Economic analyses will be published elsewhere.

**Figure 2 ijerph-19-17088-f002:**
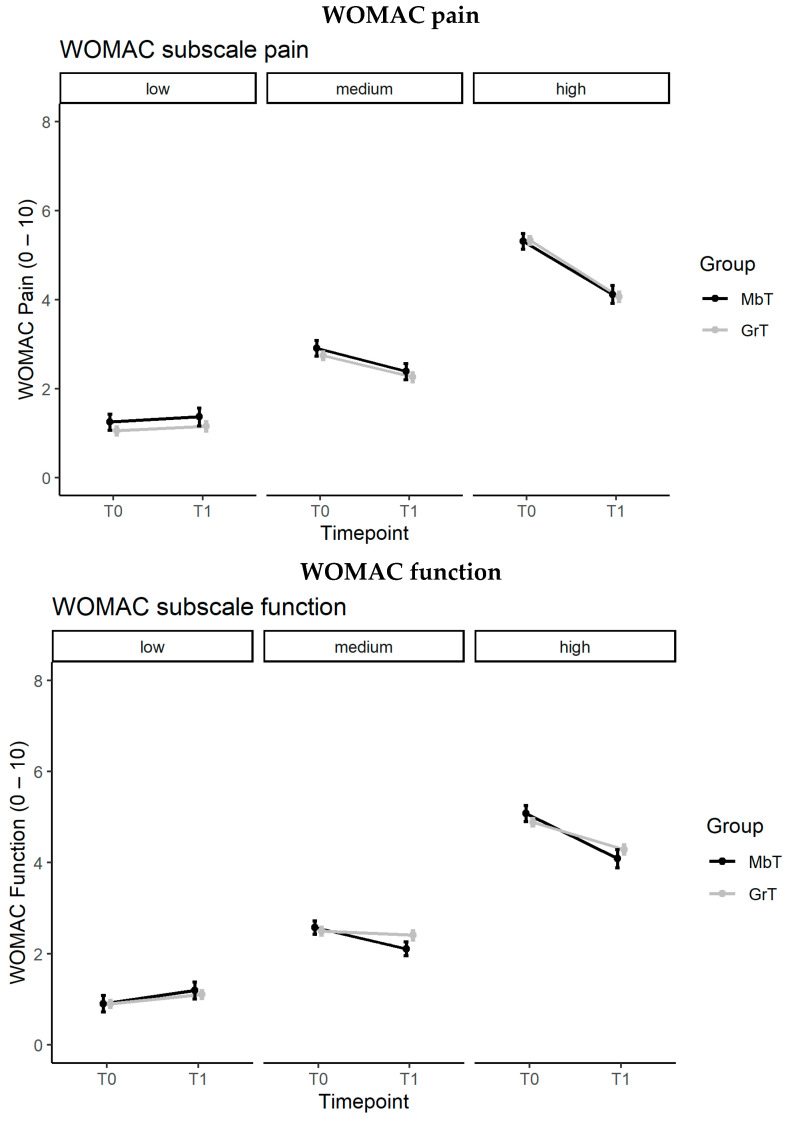
Estimated marginal means (SE) WOMAC pain/function scores in GHT and MbT at baseline T0 and post-intervention T1. GHT = Group-/home-based training, MbT = Machine-based training. Adjusted Models (Time, Treatment, Time*Treatment, Age, Sex, BMI, OA site, Artificial joint replacement). Lower scores represent better health.

**Table 1 ijerph-19-17088-t001:** Exercise Interventions.

	GHT–Group-/Home-Based Training	MBT–Machine-Based Training
**Duration**	**11-Week**	**12-Week**
**Number of Sessions**	8 supervised (1/w); 22 unsupervised home sessions (2/w)	24 supervised sessions (2/w) + strength test
**Delivery Mode**	Group sessions with max. 12 participants	First 3 sessions: 1:1 individual (strength test + introduction to machines) Following sessions: 1–2 health professionals assisting 3–10 participants
***Exercise Type***/ **Training Elements**	*Mixed*: -Mobilization, stretching and motor learning -Resistance training for hip and thigh muscles -Postural control/balance	*Strength*: -Ergometer warm-up (5 min) -Resistance training -30 s stretches for loaded muscle groups
**Type of Resistance**	-Therapeutic elastic bands + other small training devices (i.e., gymnastic ball, stability pads) -Body weight	-Weight machines -Body weight + small training devices (rubber band, gymnastic ball, weight cuffs)
**Intensity and Structure**	Progressive concept: Week 1–3: Mobilization/motor learning Week 4–7: Muscular endurance + postural control (balance static) Week 8–11: Strength + postural control (balance dynamic)	Progressive concept: Week 1–2: Motor learning (30% of MVC) Week 3–5: Muscular endurance (50% of MVC) Week 6–12: Strength (75% of MVC)
**Additional material**	Book including all home-based exercises as well as information on OA and exercise and a training log	
**Scheduled appointments**	Fixed	At customer’s option

w = week; MVC = maximum voluntary contraction.

**Table 2 ijerph-19-17088-t002:** Baseline characteristics of the study population (n = 657) by intervention groups.

	Total n = 657	GHT n = 521	MbT n = 136	*p*
** *Socio-demographics* **				
**Sex** (n = 657)				**<0.001** ^&^
Female	469 (71.4%)	397 (76.2%)	72 (52.9%)	
Male	188 (28.6%)	124 (23.8%)	64 (47.1%)	
**Age** (years) (n = 657)				**<0.001** ^$^
Mean (SD)	62.6 (10.3)	63.6 (9.5)	58.8 (11.9)	
**BMI** (kg/m²) (n = 635)				0.379 ^$^
Mean (SD)	27.9 (4.7)	27.8 (4.6)	28.2 (5.1)	
Missing	22 (3.3%)	19 (3.6%)	3 (2.2%)	
** *Anamnesis Joints* **				
**OA Affected Joint** (n = 649)				0.120 ^&^
Knee	341 (52.5%)	260 (50.5%)	81 (60.4%)	
Hip	146 (22.5%)	121 (23.5%)	25 (18.7%)	
Both	162 (25.0%)	134 (26.0%)	28 (20.9%)	
Missing	8 (1.2%)	6 (1.2%)	2 (1.5%)	
**Additional Joint Replacement at another site (hip/knee)** (n = 646)				<0.500 ^&^
Yes	81 (12.5%)	67 (13.1%)	14 (1.4%)	
No	565 (87.5%)	445 (86.9%)	120 (89.5%)	
Missing	11 (1.7%)	9 (1.7%)	2 (1.5%)	

GHT = Group-/homebased training, MbT = Machine-based training. Interquartile range (IQR), Standard Deviation (SD), ^$^ = Student’s t-test; ^&^ = Chi-squared test.

**Table 3 ijerph-19-17088-t003:** Descriptions of the WOMAC pain and function scores at T0, T1.

	Baseline (T0)			Post-Intervention (T1)	
	n	Mean (SD)		n	Mean (SD)
**WOMAC pain (0–10) ^a^**					
GHT	521	3.12 (1.98)		368	2.44 (1.87)
MbT	136	3.23 (1.92)		110	2.59 (1.73)
**WOMAC function (0–10) ^a^**					
GHT	521	2.75 (1.92)		374	2.48 (1.87)
MbT	136	2.89 (1.88)		114	2.35 (1.60)

GHT = Group-/homebased training, MbT = Machine-based training; SD = standard deviation; WOMAC: Western Ontario and McMaster Universities Osteoarthritis Index; ^a^ Lower scores represent better health.

**Table 4 ijerph-19-17088-t004:** Results for the primary outcomes WOMAC pain and function of the linear mixed models.

	Model 1P: WOMAC Pain				Model 1F: WOMAC Function			
*Variables*	numDF	denDF	F	*p*	numDF	denDF	F	*p*
**Fixed Effects**								
Time	1	523.78	63.37	**<0.001**	1	519.22	19.92	**<0.001**
Treatment	1	600.13	1.40	0.237	1	597.08	0.05	0.828
BL Severity	2	601.18	392.01	**<0.001**	2	597.65	429.56	**<0.001**
Time*Treatment	1	523.99	0.02	0.884	1	519.51	3.38	0.067
Time*BL Severity	2	523.37	31.06	**<0.001**	2	519.05	22.15	**<0.001**
Treatment*BL Severity	2	601.68	0.29	0.748	2	596.47	0.26	0.770
Time*Treatment*BL Severity	2	523.62	0.06	0.945	2	518.89	1.57	0.209
Age	1	628.01	5.29	**0.022**	1	626.78	5.46	**0.020**
Sex	1	600.92	0.15	0.702	1	600.40	0.03	0.856
BMI	1	601.76	3.49	0.062	1	599.40	7.61	**0.006**
OA site	2	619.98	0.55	0.575	2	614.50	1.83	0.161
Artificial joint	1	603.26	0.00	0.976	1	603.56	1.49	0.223
**Random Effects (SD)**								
*σ^2^*	0.77				0.62			
*τ00 id*	0.52				0.49			
*AIC*	3272.01				3116.97			
*Marginal/Conditional R^2^*	0.658/0.795				0.686/0.824			
*ICC*	0.40				0.44			

numDF = numerator degrees of freedom, denDF = denominator degrees of freedom; ICC = intraclass correlation coefficient; AIC = Akaike information criterion. BL = baseline. Note: Satterthwaite method for degrees of freedom.

**Table 5 ijerph-19-17088-t005:** Estimated marginal means (95% CI) from the LMMs and within-group change from baseline (cfb).

	Baseline (T0)	Post-Intervention (T1)	cfb	*p*
**Model 1p: WOMAC pain (0–10) ^a^**				
**Low Baseline Severity**				
**GHT (n = 160)**	1.05 (0.86–1.24)	1.16 (0.95–1.36)	+0.102	0.999
**MbT (n = 42)**	1.25 (0.89–1.61)	1.37 (0.97–1.76)	+0.117	0.999
**Medium Baseline Severity**				
**GHT (n = 183)**	2.75 (2.57–2.92)	2.26 (2.05–2.47)	−0.488	**<0.001**
**MbT (n = 46)**	2.91 (2.56–3.25)	2.39 (2.02–2.75)	−0.520	**0.032**
**High Baseline Severity**				
**GHT (n = 178)**	5.33 (5.15–5.51)	4.06 (3.85–4.27)	−1.268	**<0.001**
**MbT (n = 48)**	5.31 (4.97–5.65)	4.12 (3.73–4.51)	−1.192	**<0.001**
**Model 1f: WOMAC function (0–10) ^a^**				
**Low Baseline Severity**				
**GHT (n = 186)**	0.90 (0.74–1.06)	1.11 (0.93–1.28)	+0.208	0.544
**MbT (n = 37)**	0.90 (0.55–1.26)	1.20 (0.83–1.57)	+0.292	0.100
**Medium Baseline Severity**				
**GHT (n = 156)**	2.50 (2.33–2.67)	2.40 (2.20–2.61)	−0.094	0.999
**MbT (n = 58)**	2.58 (2.29–2.86)	2.11 (1.80–2.41)	−0.466	**0.016**
**High Baseline Severity**				
**GHT (n = 179)**	4.88 (4.72–5.04)	4.29 (4.09–4.48)	−0.596	**<0.001**
**MbT (n = 41)**	5.07 (4.72–5.42)	4.10 (3.70–4.47)	−0.982	**<0.001**

LMM = Linear mixed models, ^a^ Lower scores represent better health; cfb = predicted mean change from baseline within groups. *p*-values were Bonferroni adjusted to account for post-hoc testing with respect to the two outcomes and the two groups within each level of baseline severity.

## Data Availability

The raw datasets generated by the insurance company AOK Baden-Wuerttemberg are not publicly available due to data protection reasons.

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
