# Peer review of "Comparison of a Group-/Home-Based and a Weight-Machine-Based Exercise Training for Patients with Hip or Knee Osteoarthritis—A Secondary Analysis of Two Trial Interventions in a Real-World Context"

_ijerph, 2022, doi:10.3390/ijerph192417088_

Round 1

Reviewer 1 Report

1. The authors claim "disproportionately high use of MbT by men". However, MbT was only given for a subject choice at 2 of 70 study sites, while only GHT was offered at rest. It can be misleading information to the readers by not providing more detailed study population information at all sites, i.e. the sites offering GHT may have had a lower number of male subjects. 

In addition, the authors claim, "Females seem to participate more often in group-based exercise interventions for hip and knee OA in comparison to men" (line 446). This is also misleading if only GHT was offered at 68 sites and the choice of the subjects was only given at 2 sites. The percentage of female subjects registered for MbT was higher than that of men (52.9% vs 47.1%). 

2. The authors introduced baseline disease severity subgroups of WOMAC pain and function by low, medium, and high, based on the study by Weigl and Angst. However, Weigl and Angst study is two years follow-up study, and the current study is 11 weeks and 12 weeks short-term study. Can the authors provide further support for these terciles or the effect that could have been introduced in the comparison, potentially a limitation?

3. Regarding the subgroup terciles, can authors present the sample size that was allocated in each severity group? 

4. The authors explain that perceived benefit "is not only related to pain and functional outcomes." (line 406) This connects to the idea that MbT has a higher proportion of high to very high benefit than GHT's, thus MbT is slightly favored for an OA exercise regimen from the two. Can the authors provide correlations of the perceived benefit to the primary outcome measure?

5. Regarding the sentence in line 426 and the supplementary table 2, it is a misleading percentage calculation as 21% of men and 79% of women dropped out from the exercise programs. The percentage has to be based on the number of subjects included in the study per gender if need to be compared between the rates. For male, 35/188 (18.6%) and for female 132/469 (28.1%). And the same for the rest of the supplementary table 2 if any comparison is made.

6. The paragraph starting line 261 and the following paragraph starting line 266: Please be consistent with the digits. (cfb value and p-value, 0.999 vs .999) 

7. Cited references 16 and 18 are the same.

Reviewer 2 Report

Comparison of a GHT and a MbT exercise training for patients with hip or knee OA

Summary 

The authors compare results regarding pain and function in OA patients after a weight-machine based strengthening program (MbT) or a group-/homebased training (GHT).

They demonstrate, that both MbT and GHT showed positive results for patients with at least moderate hip and knee OA symptoms with a slight advantage for MbT. However, this was characterized as not statistically significant.

The topic is not completely new, there are similar studies in this field. The results are not really spectacular (but interesting) and the hypothesis (MbT is better than GHT) was not proven.

The study is well written with an adequate structure/study design. However, the group sizes are different (566 vs. 141), in addition follow-up examination was 10% longer (11 vs. 12 weeks) for the MbT group. Especially because of the very short overall FU, this circumstance has probably an effect on the outcome. At least, both should be mentioned as limitations of the study.

Round 2

Reviewer 1 Report

Dear authors,

Thank you for addressing the comments, and edits on the manuscript and kind explanations.